# MITIGATING OBJECT HALLUCINATIONS IN LARGE VISION-LANGUAGE MODELS VIA MULTI-SCALE VISUAL INTEGRATION

## ABSTRACT

Large Vision-Language Models (LVLMs) often suffer from object hallucination, where the model generates descriptions of objects that do not exist in the image. Our analysis reveals a boundary effect in scaling resolution of visual input: moderate increases in visual resolution alleviates hallucination, while excessive tokens diffuse attention and reintroduce errors, analogous to long-context issues in text generation. To address this, we propose a multi-scale visual decoding framework that integrates fine-grained grounding and global attention constraints, keeping captions faithful to both object-level evidence and overall semantics. A fusion-based scoring mechanism further guides decoding to suppress hallucinated objects while reinforcing faithful ones. Experiments across multiple benchmarks demonstrate that our approach effectively mitigates hallucination and delivers superior caption quality compared to state-of-the-art methods.

## 1 INTRODUCTION

Large Vision-Language Models (LVLMs) Dai et al. (2023); Bai et al. (2025) have recently demonstrated strong capabilities in visual understanding and multimodal reasoning, largely by leveraging the powerful generalization abilities of large language models (LLMs) such as Qwen Bai et al. (2023), LLaMA Touvron et al. (2023), and GPT Brown et al. (2020). Despite these successes, LVLMs still suffer from object hallucination, where models generate descriptions of non-existent objects or fabricated details Zhai et al. (2024); Hu et al. (2023). This issue severely undermines their reliability in applications that demand precise visual-textual grounding.

Recent studies have attempted to mitigate hallucination through mostly training-free decoding strategies. Some works attribute hallucination to the LLM decoding process and design contrastive or constrained decoding methods to balance linguistic fluency with visual grounding Leng et al. (2023); Manevich & Tsarfaty (2024); Kim et al. (2024). Others recalibrate model focus via attention modulation Huang et al. (2024); Chuang et al. (2024). While effective to some extent, these approaches mainly intervene at the decoding stage, without fundamentally addressing the limitations of the underlying visual inputs that provide grounding evidence.

We argue that hallucination in LVLMs often stems from two sources: insufficient fine-grained grounding in the visual encoder and contextual bias in the LLM decoder. On the one hand, existing visual encoders compress images into tokenized representations that frequently lose fine-grained details due to resolution bottlenecks Li et al. (2023); Chen et al. (2024). On the other hand, during autoregressive decoding, LLMs tend to drift away from visual evidence as textual context accumulates, relying increasingly on contextual plausibility rather than image grounding Zhou et al. (2024); Wang et al. (2024a). Existing models typically attempt to mitigate hallucination by increasing input resolution, allowing the encoder to capture more visual details. However, our analysis shows that this strategy faces an inherent upper bound: as resolution increases, the number of visual tokens grows excessively, leading to an "attention diffusion" effect where the model struggles to focus on relevant regions. To verify this, we conduct controlled experiments across different resolutions and observe a boundary effect—moderate enrichment of visual information reduces hallucination, but excessively high resolutions cause performance degradation, with hallucinations reemerging.

Figure 1: (a) The trend of hallucination rates on the CHAIR Rohrbach et al. (2019) benchmark as resolution increases. Qwen2.5-VL-7B Bai et al. (2025) is selected as backbone. Below 512, higher resolution consistently reduces hallucination; however, beyond 512, hallucination rises again with further resolution scaling. (b) An example of VQA question reformed from CHAIR benchmark

Motivated by these findings, we propose a multi-scale visual decoding framework to address the boundary effect observed in resolution scaling. Instead of simply increasing resolution, our approach introduces fine-grained grounding to recover lost object-level details and incorporates global attention constraints to maintain focus on the most relevant regions. By jointly enforcing these complementary signals, the model can exploit detailed evidence without being distracted by redundant tokens. A fusion module further integrates grounding confidence, attention constraints, and LVLM likelihood scores to guide decoding. This design effectively mitigates hallucination while requiring no additional training or external supervision.

Overall, our contributions are summarized as follows:

1. We revisit the origins of object hallucination in LVLMs, focusing on the limitations of low-resolution visual encoders and the decoding process of LLMs. Through carefully designed experiments, we validate our analysis and show that moderate resolution scaling mitigates hallucination, while an analogous long-context hallucination also emerges in the visual domain when visual tokens become overly long under high resolution.

2. We propose a novel decoding framework that integrates fine-grained grounding from grounding visual inputs with global semantic fidelity from visual attention inputs. By combining these visual supplements with the likelihood scores of LVLMs, our method mitigates hallucination while preventing the attention diffusion caused by excessively visual tokens, ensuring that the model remains focused on the correct visual regions.

3. Experimental results validate the effectiveness of our method, demonstrating significant performance improvements across multiple benchmarks while also demonstrating superior generation quality.

## 2 VISUAL RESOLUTION EFFECTS ON HALLUCINATION

We argue that a key factor underlying hallucination is the limited capacity of visual encoders to capture fine-grained object-level information. When such detailed grounding is absent, the model increasingly relies on language biases, which amplifies hallucination generation. Building on this, we conduct extensive experiments to validate our analysis.

Early studies such as Li et al. (2024) have suggested that the resolution of the visual encoder plays a crucial role in mitigating hallucination. Our results indicate that recent LVLMs still suffer from strong resolution dependence. Using the CHAIR Rohrbach et al. (2019) benchmark, we constructed a hallucination-oriented VQA task with an encoder initialized at a low resolution as illustrated in Figure 1b. Further details are provided in Appendix B. We then progressively scaled the input resolution and tracked the corresponding changes in hallucination rates, as illustrated in Figure 1a.

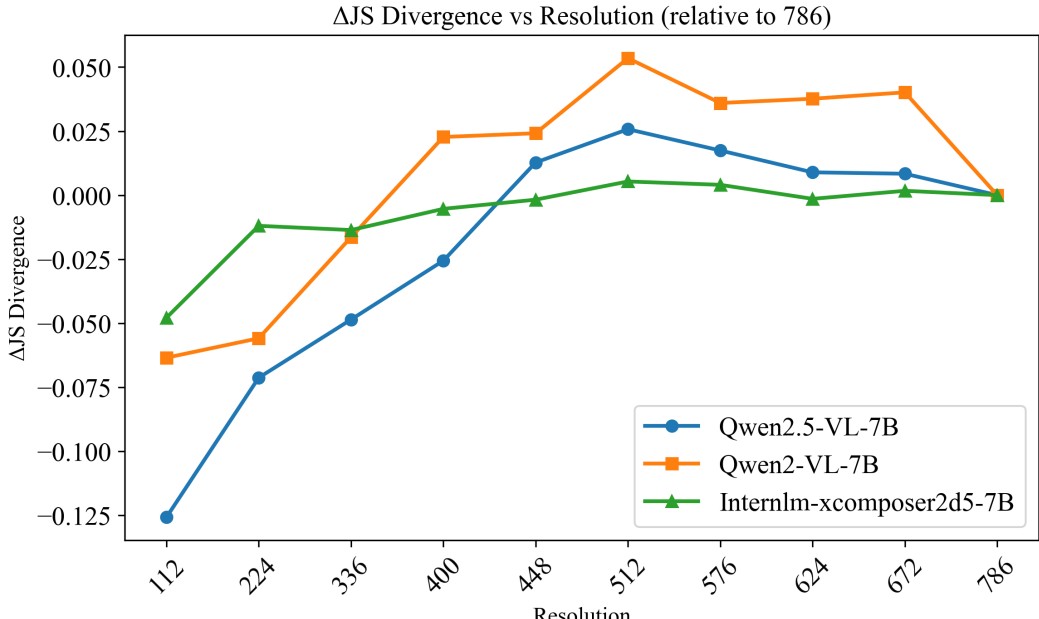

Figure 2: We compare the logit similarity of hallucinated tokens across different resolutions with various backbone. At 512, the logits diverge most from the language prior (resolution = 0), indicating the strongest contribution of visual evidence. As resolution continues to grow, the logits gradually regress toward the language prior, reflecting reduced visual grounding.

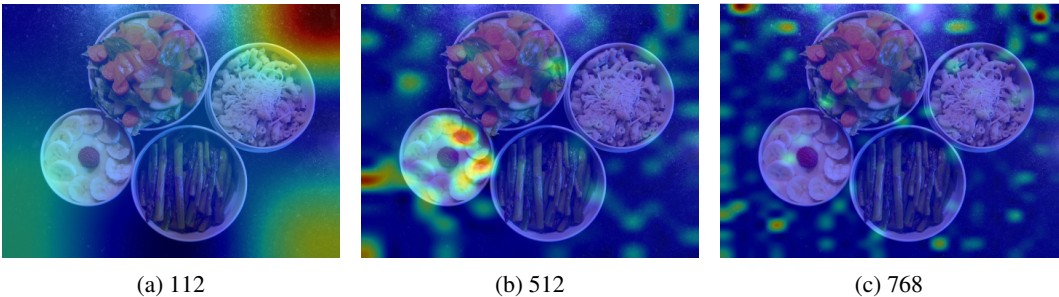

(a) 112                    (b) 512                    (c) 768

Figure 3: Visualization of attention maps under different resolution settings

**Boundary Effect of Resolution Scaling and Logit Similarity Analysis:** We observe a boundary effect in resolution scaling: while moderate increases significantly reduce hallucination, overly high resolutions lead to performance degradation, with hallucinations reemerging rather than continuing to decline. To further analyze the interaction between vision and language, we employ multiple models Dong et al. (2024); Bai et al. (2025); Wang et al. (2024b) to compute the distribution of logits across different resolutions on VQA tasks. We additionally treat the case of resolution = 0 as representing the language-only prior of the LLM, since no visual input is provided and the model generates token solely on textual context to answer the VQA questions. We then measure the Jensen–Shannon similarity between model outputs and this language-only prior (Figure 2) as shown in 1 and 2.

$$y_t \sim \text{softmax}\big(f_\theta(y_t \mid v_{resolution}, x, y_{<t})\big), \tag{1}$$

$$\text{JS}(r_i, r_j) = \text{JSD}\Big(y_t^{(r_i)} \,\|\, y_t^{(r_j)}\Big), \tag{2}$$

where $v_{resolution}$ denotes the visual inputs under different resolutions, $x$ represents the input context, and $y_{<t}$ refers to the previously generated tokens up to step $t$.

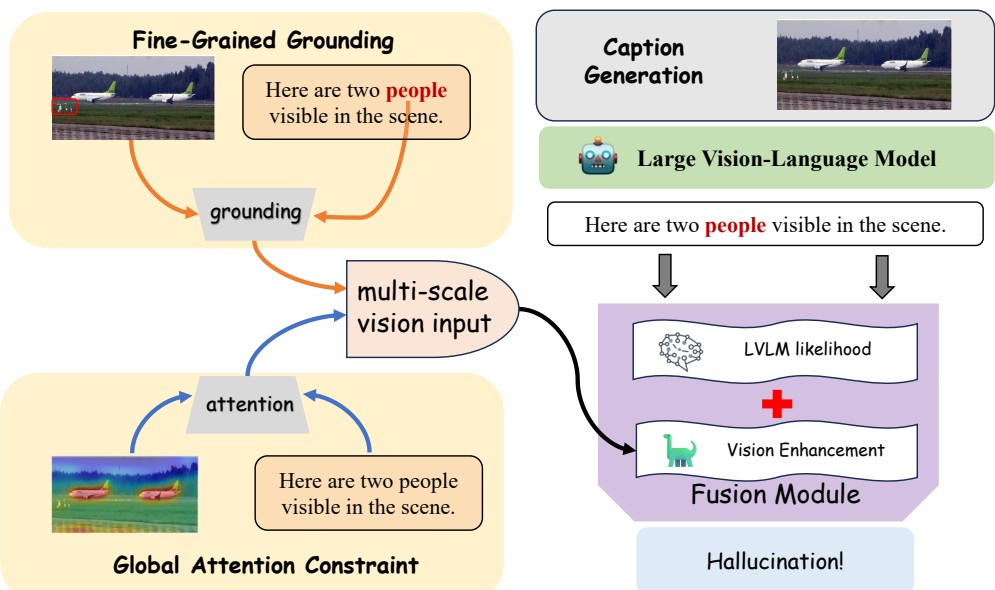

Figure 4: Overview of our method: We propose a multi-scale visual decoding framework that jointly enforces fine-grained and global attention constraints to guide LVLM generation. By leveraging these complementary constraints, the framework ensures that the model attends to the correct visual regions, effectively mitigating hallucinations while preserving overall semantic consistency. Candidate outputs are iteratively refined under this framework, resulting in captions that are both faithful to object-level evidence and coherent at the global level.

The divergence peaks at an intermediate resolution, indicating that visual evidence contributes most effectively at this scale. Beyond this point, the similarity gradually decreases, suggesting that model generation drifts back toward language priors despite higher-resolution inputs. As noted in prior work Zhou et al. (2024); Wang et al. (2024a), hallucination severity tends to increase with longer textual sequences, as models overemphasize local continuity while disregarding the input signal. We identify an analogous effect in the visual domain: as resolution increases, the growing number of visual tokens disperses attention, weakening effective grounding and reintroducing language-driven hallucination. Figure 3 illustrates this phenomenon, showing that at excessively high resolutions, attention becomes broadly distributed across the image rather than concentrated on salient regions, thereby undermining grounding.

**Hypothesis.** While higher resolution alleviates hallucination by enriching visual information, resolution scaling alone is insufficient. Effective mitigation requires mechanisms that guide attention to the correct regions, allowing the model to leverage fine-grained details without being overwhelmed by redundant tokens.

To address this, we propose a multi-scale visual input framework that integrates fine-grained grounding with global attention constraints, achieving more effective hallucination mitigation. Extensive experiments validate our hypothesis, demonstrating that the proposed framework significantly achieves both visual grounding and global attention constraint.

## 3 METHOD

In this section, we present the detailed approach of our method. An overview of the framework is shown in Figure 4. Our method incorporates multi-scale visual inputs, where high-resolution inputs contribute fine-grained grounding, while lower-resolution features provide global attention constraint. A fusion module then integrates these complementary information to iteratively refine the generated captions, ensuring both global consistency and object-level grounding.

## 3.1 FINE-GRAINED GROUNDING

To address the limitation of visual encoders in capturing fine-grained object details, we integrate *GroundingDINO* as a high-resolution grounding module. GroundingDINO localizes fine-grained object regions and aligns them with the corresponding caption tokens, thereby ensuring that each described object is supported by concrete visual evidence. This object-level grounding complements the global consistency provided by lower-resolution inputs, reinforcing faithfulness and reducing hallucination at a finer scale.

At time step $i$, given an image $x_{\text{img}}$ and a set of sampled sentences $\{s_{i1}, s_{i2}, \ldots, s_{ik}\}$, we apply named entity recognition (NER) Neumann et al. (2019) to extract object terms in sampled sentences. For each extracted object term $w_{ijl}$, we compute a confidence score $\sigma_{ijl}$ from the transformer attention maps Liu et al. (2024):

$$\sigma_{ijl} = \max_{w_{ijl}}(p_{ij,w_{ijl}}), \tag{3}$$

and define the grounding score as:

$$G(x_{\text{img}}, w_{ijl}) = \begin{cases} C, & \text{if } \sigma_{ijl} > \delta, \\ 0, & \text{otherwise,} \end{cases} \tag{4}$$

where $\delta$ is the threshold from the default GroundingDINO settings, and the hyperparameter $C$ denotes the weight of the grounding score. This score indicates how well the object is represented in the image.

For each sentence $s_{ij}$, we aggregate word-level scores into a sentence-level grounding score:

$$G(x_{\text{img}}, s_{ij}) = \min_{w_{ijl}} G(x_{\text{img}}, w_{ijl}), \tag{5}$$

where $\min_{w_{ijl}}$ denotes the minimum value of the grounding score across all $w_{ijl}$ in $s_{ij}$. The aggregated score gives a measure of how well the sentence reflects the visual content of the image, particularly with regard to the objects mentioned.

The rationale for using a constant instead of a likelihood score, and for adopting the minimum rather than the average or maximum, is detailed in Appendix E. This high-resolution grounding penalizes hallucinated objects while reinforcing valid ones mentioned in the caption.

## 3.2 GLOBAL ATTENTION CONSTRAINT

While high-resolution grounding enforces object-level fidelity, it is also crucial to maintain attention between the generated caption and the image to avoid the effect of overextended tokens. To this end, we employ the CLIP module at lower resolution, which embeds both text and images into a shared semantic space. This allows us to measure global image-text alignment and introduce global attention constraint.

At time step $i$, given an image $x_{\text{img}}$ and a set of sampled sentences $\{s_{i1}, s_{i2}, \ldots, s_{ik}\}$, we compute a sentence-level alignment score:

$$A^{sentence}(x_{\text{img}}, s_{ij}) = \cos\left(f_{\text{CLIP}}(x_{\text{img}}), f_{\text{CLIP}}(s_{ij})\right), \tag{6}$$

where $f_{\text{CLIP}}(\cdot)$ denotes the CLIP embedding function. A higher cosine similarity indicates stronger alignment between the sentence and the image.

To further constrain hallucination at the object level, we apply named entity recognition (NER) Neumann et al. (2019) to extract object terms (e.g., "dog", "ball"), and compute word-level alignment:

$$A^{word}(x_{\text{img}}, w_{ijl}) = \cos\left(f_{\text{CLIP}}(x_{\text{img}}), f_{\text{CLIP}}(w_{ijl})\right). \tag{7}$$

The final alignment score is a weighted combination of sentence- and word-level alignment:

$$A(x_{\text{img}}, s_{ij}) = \gamma \cdot A^{sentence}(x_{\text{img}}, s_{ij}) + (1 - \gamma) \cdot A^{word}(x_{\text{img}}, s_{ij}), \tag{8}$$

where $\gamma$ balances global semantic alignment and object-level grounding at the intermediate resolution scale.

| Method | InstructBLIP | | | mPLUG-Owl2 | | | LLAVA-1.5 | | |
|---|---|---|---|---|---|---|---|---|---|
| | $CHAIR_S\downarrow$ | $CHAIR_I\downarrow$ | $BLEU\uparrow$ | $CHAIR_S\downarrow$ | $CHAIR_I\downarrow$ | $BLEU\uparrow$ | $CHAIR_S\downarrow$ | $CHAIR_I\downarrow$ | $BLEU\uparrow$ |
| Greedy | 57.9 | 17.1 | 15.9 | 52.7 | 16.0 | 18.1 | 47.0 | 13.6 | 18.9 |
| DoLa | 55.6 | 17.0 | 16.5 | 52.6 | 15.2 | 18.1 | 46.6 | 13.6 | 19.2 |
| VCD | 63.2 | 19.5 | 17.7 | 51.4 | 16.0 | 17.5 | 44.6 | 12.5 | 17.8 |
| OPERA | 51.5 | 15.6 | 18.3 | 48.5 | 16.1 | 17.9 | 49.5 | 13.7 | 18.4 |
| HALC | 61.6 | 18.9 | 18.1 | 51.7 | 15.5 | 17.4 | 40.6 | 11.0 | 19.0 |
| AGLA | 49.0 | 12.1 | 16.8 | 47.6 | 12.0 | 19.0 | 43.0 | 14.1 | 18.8 |
| CODE | 37.8 | 11.1 | 16.1 | 41.7 | 12.3 | 17.4 | 35.4 | 9.3 | 18.8 |
| SID | 43.6 | 13.1 | 16.4 | 46.0 | 12.9 | **19.1** | 45.0 | 11.7 | 18.4 |
| **Ours** | **23.5** | **6.3** | **19.4** | **24.6** | **6.8** | 18.0 | **22.2** | **5.8** | **19.4** |

Table 1: Experimental results of different decoding methods on various LVLMs using the **MSCOCO-CHAIR** Lin et al. (2015) dataset. The results are reproduced based on the original papers or official code. $C_S$ refers to $CHAIR_S$, $C_I$ refers to $CHAIR_I$ and $B$ refers to BLEU-1 Score. Higher BLEU-1 scores indicate better text generation quality, while lower $CHAIR_S$ and $CHAIR_I$ scores reflect stronger hallucination mitigation. **Bold** values indicate the best performance across other methods.

### 3.3 MODULE FUSION:

To preserve the generative quality of LVLMs, we introduce the concept of LLM likelihood and utilize this probability as a scoring metric in the subsequent evaluation. Given a premise question text $s = \{t_1, t_2, \ldots, t_m\}$, where $t_i$ denotes the token generated at the $i$-th timestep, we utilize Predictive Entropy (PE) for uncertainty estimation Kadavath et al. (2022); Duan et al. (2024); Kuhn et al. (2023), which is defined as the entropy of the entire sentence. To mitigate the impact of generation length on predictive entropy and ensure the proper functioning of LVLM, we adopt a variant known as length-normalized predictive entropy as Equation 9. This variant divides the joint log-probability of each sequence by the length of the sequence, as proposed by Malinin and Gales Malinin & Gales (2021) in the context of natural language generation (NLG) uncertainty, and has been empirically shown to be advantageous in the work by Kuhn Kuhn et al. (2023).

$$f_\theta(s_i) = \frac{1}{m} \sum_{i=1}^{m} -\log p_\theta(x_i|s_{<i}) \tag{9}$$

where $\theta$ represents the LVLM parameters and $m$ is the length of the generated sentence.

To combine the strengths of both the grounding and constraint inputs, we propose a score fusion strategy. This module integrates the consistency score from global attention constraint, the visual alignment score from grounding, and the internal likelihood score from the LVLM to compute a final fusion score. The final score $F(x_{\mathrm{img}}, s_{ij})$ for each generated caption $s_{ij}$ is given by:

$$F(x_{\mathrm{img}}, s_{ij}) = (1-\alpha)\big(G(x_{\mathrm{img}}, s_{ij}) + A(x_{\mathrm{img}}, s_{ij})\big) + \alpha \cdot f_\theta(s_{ij}) \tag{10}$$

where $G_{\mathrm{fine}}(x_{\mathrm{img}}, s_{ij})$ is the fine-grained visual grounding score from the grounding module, $A(x_{\mathrm{img}}, s_{ij})$ is the global attention score capturing overall image context, and $f_\theta(s_{ij})$ is the internal likelihood from the LVLM for caption $s_{ij}$. The hyperparameter $\alpha$ controls the influence of these auxiliary visual scores on the decoding distribution; when $\alpha = 1$, the fusion module reduces to standard greedy decoding.

The fusion score $F(x_{\mathrm{img}}, s_{ij})$ provides a comprehensive measure of hallucination by integrating both fine-grained and global attention constraint. It is used to rank generated captions, with top candidates selected for further refinement. We adopt an iterative decoding strategy to progressively improve caption quality. At each iteration, the LVLM generates a set of candidate captions, which are evaluated using the fusion score $F(x_{\mathrm{img}}, s_{ij})$. The top $N$ candidates are retained and refined over $k$ iterations, ensuring that captions gradually leverage both object-level grounding and global attention to enhance faithfulness and visual fidelity. The full iteration procedure is summarized in Algorithm 1 (Appendix F).

## 4 EXPERIMENTS

In this section, we evaluate the performance of our method on caption generation, focusing on its effectiveness in mitigating object hallucination while maintaining caption quality. Our experiments include CHAIR, OPOPE, and GPT-4V-assisted evaluations. Additional experimental results and analyses are provided in Appendix B.

| Setting | Decoding | InstructBLIP | | | mPLUG-Owl2 | | | LLAVA-1.5 | | |
|---|---|---|---|---|---|---|---|---|---|---|
| | | $A \uparrow$ | $P \uparrow$ | $F_{0.2} \uparrow$ | $A \uparrow$ | $P \uparrow$ | $F_{0.2} \uparrow$ | $A \uparrow$ | $P \uparrow$ | $F_{0.2} \uparrow$ |
| Random | Greedy | **76.8** | 94.2 | **91.8** | **75.1** | 92.3 | 90 | **78.4** | 94.8 | 92.6 |
| | HALC | 76.7 | 93.8 | 91.5 | 73.7 | 92.2 | 89.5 | 73.8 | 95.8 | 92.4 |
| | Ours | 71.9 | **94.4** | 90.8 | 72.3 | **95.0** | **91.4** | 73.4 | **96.1** | **92.6** |
| Popular | Greedy | 73.1 | 83.9 | 82.4 | **71.5** | 82.6 | 81 | **74.9** | 85.7 | 84.3 |
| | HALC | **73.3** | 84.2 | 82.7 | 70.2 | 82.1 | 80.3 | 71.4 | 87.7 | 84.9 |
| | Ours | 70.1 | **87.9** | **84.9** | 70.5 | **88.8** | **85.8** | 72.7 | **90.3** | **87.6** |
| Adversarial | Greedy | **72.5** | 82.6 | 81.2 | 68.9 | 76.5 | 75.3 | **73.1** | 81.5 | 80.4 |
| | HALC | 71.2 | 79.4 | 78.2 | 68.4 | 77.5 | 76.1 | 70.3 | 84.6 | 82.2 |
| | Ours | 69.2 | **85.0** | **82.4** | **68.9** | **83.5** | **81.1** | 70.7 | **86.9** | **84.3** |

Table 2: Experimental results of different decoding methods on various LVLMs in the **MSCOCO-OPOPE** Chen et al. (2024). The results are reproduced using the original papers or official code. A refers to Accuracy, P refers to Precision and $F_{0.2}$ refers to $F_{0.2}$ Score. Higher Accuracy, Precision and $F_{0.2}$ Score scores indicate better quality, whereas lower $CHAIR_S$ and $CHAIR_I$ scores reflect stronger hallucination mitigation. **Bold** values represent the best results among all methods.

## 4.1 EXPERIMENT SETUPS

**Baselines:** To effectively evaluate our method, we include regular greedy decoding as baselines. Additionally, we incorporate state-of-the-art methods specifically designed to mitigate object hallucination (OH), including DoLa Chuang et al. (2024), OPERA Huang et al. (2024), VCD Leng et al. (2023), CODE Zhai et al. (2024) ALGA An et al. (2025), HALC Chen et al. (2024) and SID Huo et al. (2024) in our analysis.

**LVLM Backbones:** We conduct our experiments on different LVLMs—InstructBLIP Dai et al. (2023), LLaVA-1.5 Liu et al. (2023), and mPLUG-Owl2 Li et al. (2022)—to evaluate our method and all the previously mentioned baselines.

## 4.2 METRICS

**Datasets:** We conduct our experiments mainly on three benchmark:We conducted our experiments primarily on three benchmarks: CHAIR, POPE, and a GPT-4V assisted evaluation. Detailed descriptions of these datasets are provided in Appendix A. The results across all three benchmarks consistently demonstrate the effectiveness of our approach in mitigating object hallucination, while maintaining high-quality text generation.

**CHAIR:** To evaluate the effectiveness of our method in mitigating object hallucination, we follow the standard CHAIR evaluation setting Rohrbach et al. (2019). For all backbones, we use the prompt "Please describe this image in detail" . The generated parameters for our method are provided in Appendix B, with hyperparameters $\alpha = 0.01$ and $\gamma = 0.5$. The CHAIR results are shown in Table 1. Throughout the experiments, our method achieves state-of-the-art (SOTA) performance in reducing hallucinations across other methods while maintaining caption generation quality. Specifically, our method achieves improvement over the previous SOTA under the CHAIR metrics. We observe that our method performs better with backbones exhibiting high levels of hallucination, which can be attributed to the alignment module's effectiveness in mitigating hallucinations. The generated sentences contain an average of 80 to 90 words, with the max new tokens parameter set to 512. Notably, we conduct experiments on different lengths of generated captions in Appendix and evaluate our method on other LVLMs under the CHAIR benchmark in Appendix. These results demonstrate that the superior performance of our method remains consistent across both long and short description generation tasks.

**POPE:** Following the HALC's OPOPE setup Chen et al. (2024), We conduct the POPE experiment, and the results are presented in Table 2. Throughout the evaluation, our method achieves better results compared to the greedy baseline and the HALC method, despite yielding a lower accuracy score. As noted by HALC, false positives become less reliable in offline POPE testing, and the diversity of described content may introduce biases in true positive samples. Consequently, this can result in deviations in the accuracy metric. Therefore, we primarily utilize precision and $F_{0.2}$ score as reference metrics. According to our experimental results, our approach achieves state-of-the-art performance within HALC's OPOPE framework.

| Method | InstructBLIP | | mPLUG-Owl2 | | LLAVA-1.5 | |
|--------|------|------|------|------|------|------|
| | C | D | C | D | C | D |
| Greedy | 4.47 | 5.11 | 5.10 | 5.71 | 5.95 | 6.11 |
| **Ours** | **5.08** | **5.93** | **5.83** | **5.74** | **6.27** | **6.34** |
| OPERA | 5.44 | 5.75 | 5.35 | 5.70 | 5.98 | 6.24 |
| **Ours** | **5.90** | **6.04** | **5.78** | **5.71** | **6.11** | **6.29** |
| HALC | 5.95 | **6.34** | 5.51 | **6.29** | 5.10 | 4.91 |
| **Ours** | **6.27** | 6.11 | **6.24** | 6.16 | **6.28** | **6.40** |

Table 3: Experimental results of different decoding methods on GPT4V assist evaluation in OPERA Chen et al. (2024).

| Parameter | $C_S$ | $C_I$ | $P \uparrow$ | $F_{0.2} \uparrow$ |
|-----------|-------|-------|------|--------|
| $\alpha = 0.9$ | 32.2 | 9.2 | 89.9 | 86.6 |
| $\alpha = 0.1$ | 25.6 | 6.6 | 90.6 | 87.0 |
| $\alpha = 0.01$ | **22.2** | **5.8** | **91.1** | **88.2** |
| $\gamma = 0.2$ | 22.4 | **5.7** | 91.1 | 88.0 |
| $\gamma = 0.5$ | **22.1** | 5.8 | **91.1** | 88.2 |
| $\gamma = 0.8$ | 22.2 | 6.0 | 91.0 | **88.3** |

Table 4: Ablation study on different values of $\alpha$ and $\gamma$.

| **Greedy** | **Relation** | **Attribute** | **Object** | $CHAIR_S$ | $CHAIR_I$ | $P \uparrow$ | $F_{0.2} \uparrow$ |
|---------|----------|-----------|--------|---------|---------|------|--------|
| ✓ | | | | 47.0 | 13.6 | 87.3 | 85.8 |
| | ✓ | | | 45.5 | 14.4 | 87.3 | 85.6 |
| | | ✓ | | 42.4 | 13.1 | 88.7 | 87.1 |
| | | | ✓ | **38.9** | **11.7** | **89.6** | **87.2** |
| | ✓ | ✓ | | 40.3 | 12.6 | 88.4 | 86.0 |
| | ✓ | | ✓ | 40.5 | 12.4 | 88.2 | 86.7 |
| | | ✓ | ✓ | 42.4 | 12.1 | 88.6 | 86.4 |
| | ✓ | ✓ | ✓ | 45.4 | 12.8 | 87.2 | 85.7 |

Table 5: Comparison of performance for Word Categories in CLIP input.

**GPT-4V assisted evaluation:** Following the OPERA Huang et al. (2024) protocol, we conduct a GPT-4V assisted evaluation to assess the effectiveness of our method in mitigating hallucinations in generated captions. Notably, we observe that GPT-4V tends to assign higher scores to captions presented second in sequence. To mitigate this bias, we conduct a second round of evaluation where the order of captions in each pair was swapped. The evaluation results, adjusted for order bias, are presented in the Table 3. And the comprehensive results of GPT-4V assisted evaluation are shown in Appendix. Experimental results demonstrate that our method outperforms existing approaches in both hallucination mitigation and generation quality.

## 4.3 ABLATION STUDY AND ANALYSIS

All ablation experiments are conducted using LLaVA-1.5 as the backbone model, with hyperparameters consistent with those described in Section 4.1 for LLaVA-1.5.

**Effectiveness of Modules:** To demonstrate the effectiveness of individual modules and the improvement in hallucination mitigation achieved by combining them, we conducted ablation experiments under four conditions, as shown in Table 6. The results demonstrate that both the GroundingDINO and CLIP modules, when used individually, significantly reduce hallucinations, with the effect being more pronounced when using grounding alone. This validates the effectiveness of using grounding and constraint as alignment mechanisms in our approach. Moreover, when both grounding and constraint are used together, the performance surpasses that of either module alone, confirming the enhanced effect of their combined supervision.

**Granularity of Inputs:** We conduct ablation experiments by using *object*, *attribute*, and *relation* as separate inputs for GroundingDINO and CLIP. Additionally, we test various input combinations for CLIP to evaluate the effectiveness of both object and sentence inputs. The greedy setting refers to the greedy decoding baseline used for comparison. The GroundingDINO experimental results are shown in Table 6 in Appendix.

Although *attribute* and *relation* are semantically important categories, and previous studies, such as HALC and HalluciDoctor Yu et al. (2024), have used *existence*, *attribute*, and *relation* as keywords for hallucination mitigation, our results indicate that the best performance in hallucination mitigation is achieved when only *object* is used as the input for CLIP, as shown in Table 5.

Our analysis reveals that grounding module struggles to effectively localize *attribute* and *relation*, resulting in excessive meaningless grounding. This issue is also discussed in R-Bench Wu et al.

| Greedy | Grounding | Object | Sentence | CHAIR$_S$ | CHAIR$_I$ | $P \uparrow$ | $F_{0.2} \uparrow$ |
|:---:|:---:|:---:|:---:|:---:|:---:|:---:|:---:|
| ✓ | | | | 47.0 | 13.6 | 87.3 | 85.8 |
| | | ✓ | | 46.2 | 12.5 | 88.2 | 86.7 |
| | | | ✓ | 44.2 | 12.4 | 87.2 | 85.7 |
| | | ✓ | ✓ | 38.0 | 11.4 | 89.7 | 87.2 |
| | ✓ | | | 24.8 | 6.8 | 90.4 | 87.2 |
| | ✓ | ✓ | ✓ | **22.2** | **5.8** | **91.1** | **88.2** |

Table 6: Comparison of performance under different input for Grounding and Constraint.

(2024). Considering the time efficiency of the grounding module, we ultimately choose to use only *object* as the input for grounding and constraint.

**Hyper Parameters:** Due to the use of multiple modules in our method, we conduct detailed ablation experiments on various hyperparameters of the model.

We first investigate the impact of the auxiliary-to-likelihood score ratio, controlled by the hyper-parameter $\alpha$ in Eq. 10. Specifically, we evaluate $\alpha \in \{0.01, 0.1, 0.9\}$ and report the corresponding $CHAIR_S$ and $CHAIR_I$ metrics in Figure 4. We then examine the balance between word-level and sentence-level CLIP scores, modulated by $\gamma$. Experiments with $\gamma \in \{0.2, 0.5, 0.8\}$ are presented in Figure 4, showing the effect of this trade-off on hallucination performance.

We observe that the model achieves the best performance when $\alpha = 0.01$. For the $\gamma$ parameter, values such as 0.2 and 0.8 lead to instability across different datasets, whereas $\gamma = 0.5$ yields more consistent results. Based on this analysis, we fix $\alpha = 0.01$ and $\gamma = 0.5$ for all experiments.

# 5 LIMITATIONS.

While our method demonstrates strong effectiveness in mitigating hallucinations, there are two primary limitations.

(1) While the method performs well across general benchmarks, its effectiveness in specialized domains, such as medical imaging, low-resource languages, or scenes with densely packed objects remains underexplored. Nonetheless, preliminary experiments in the safety domain have yielded promising results as shown in Appendix E. In future work, we plan to further validate its effectiveness across a broader range of application scenarios.

(2) Although our method remains reasonably efficient in practice, there remains room for improvement in decoding speed. We provide a detailed time complexity analysis and discuss potential acceleration strategies in Appendix C.

# 6 CONCLUSION

Motivated by our discovery of a boundary effect in visual resolution scaling—where moderate enrichment of visual information reduces hallucination, but excessive visual tokens diffuse attention and reintroduce errors—we propose a multi-scale visual decoding framework to mitigate object hallucinations in large vision-language models (LVLMs). By combining complementary fine-grained and global visual information with a fusion-based scoring mechanism, our method ensures that generated captions are grounded in both object-level grounding and gobal attention. Extensive experiments across multiple benchmarks show that our approach consistently reduces hallucinations, enhances caption quality, and achieves improvement in all metrics over state-of-the-art methods. Importantly, the effectiveness of hallucination mitigation relies on both fine-grained information and global attention constraints, and our framework maintains robust performance for both short and long descriptions without additional training or external data, making it a practical solution for existing LVLMs.

## 7 REPRODUCIBILITY STATEMENT

For the VQA hallucination and logits experiments discussed in Section 3, we provide a detailed explanation in Appendix A. The experiments reported in Sections 5.2 and 5.3 can be reproduced by following the experimental setup described in Section 5.1 and Appendix A. In addition, we present the algorithmic details of our framework in Appendix D.

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

## A  DATASETS

**CHAIR:** The Caption Hallucination Assessment with Image Relevance (CHAIR) Rohrbach et al. (2019) tool is specifically designed to assess hallucinations in image captioning tasks. It quantifies hallucinations by evaluating how many objects mentioned in the caption are absent from the ground truth label set. CHAIR provides two distinct evaluation metrics: $CHAIR_S$, which measures the proportion of hallucinated sentences relative to the total number of sentences, and $CHAIR_I$, which evaluates the proportion of hallucinated objects relative to the total number of generated objects. Lower scores on either metric indicate fewer hallucinations. We also evaluate the methods using BLEU Papineni et al. (2002), a caption-related metric that measures the similarity between generated and ground truth captions. Higher BLEU scores, specifically BLEU-1, indicate better generation quality.

**OPOPE:** Polling-based Object Probing Evaluation (POPE) is a method specifically designed to assess hallucination issues in LVLM. POPE focuses on evaluating object hallucination by utilizing an essay-style prompt in the format: "Is there a <object> in the image?" to pose visual question answering (VQA) queries to the model. The complete POPE test is divided into three splits: Random Split: Objects are randomly selected from the entire dataset for evaluation. Popular Split: This split assesses the presence of objects that most frequently appear in the dataset. Adversarial Split: This evaluates the model's ability to identify objects that are highly relevant to those present in the image.

| Method | InstuctBlip | | mPLUG-Owl2 | | LLAVA-1.5 | |
|--------|-------------|--|------------|--|-----------|--|
| | $CHAIR_S \downarrow$ | $CHAIR_I \downarrow$ | $CHAIR_S \downarrow$ | $CHAIR_I \downarrow$ | $CHAIR_S \downarrow$ | $CHAIR_I \downarrow$ |
| Greedy | 30.9 | 12.3 | 23.2 | 8.3 | 20.8 | 6.8 |
| VCD | 30.3 | 12.6 | 27.3 | 9.7 | 23.3 | 7.90 |
| OPERA | 30.0 | 11.7 | 22.1 | 7.6 | 21.1 | 6.7 |
| HALC | 30.0 | 11.4 | 17.3 | 7.4 | 13.8 | 5.5 |
| **Ours** | **21.8** | **8.1** | **16.4** | **5.9** | **11.5** | **4.2** |

Table 7: Experimental results of various methods with a 64 max new tokens setting on different LVLMs in the **MSCOCO-CHAIR** dataset. Results are reproduced using the original papers and official code.

| Method | MiniGPT-4 | | | LLAVA-Next | | |
|--------|-----------|--|--|------------|--|--|
| | $CHAIR_S \downarrow$ | $CHAIR_I \downarrow$ | $BLEU \uparrow$ | $CHAIR_S \downarrow$ | $CHAIR_I \downarrow$ | $BLEU \uparrow$ |
| Greedy | 40.6 | 14.1 | 16.7 | 19.8 | 6.2 | 16.6 |
| Nucleus | 34.0 | 12.5 | 17.3 | 23.0 | 7.9 | 16.3 |
| TopK | 35.0 | 12.5 | 17.1 | 21.2 | 7.1 | 16.4 |
| Beam | 32.2 | 11.9 | 17.1 | 15.5 | 5.5 | 16.8 |
| DoLa | 31.8 | 11.6 | 17.0 | 17.8 | 6.1 | **16.8** |
| VCD | 35.7 | 13.8 | **18.1** | 21.4 | 7.3 | 16.4 |
| OPERA | 36.4 | 12.7 | 17.0 | 17.8 | 6.1 | 16.8 |
| HALC | 34.3 | 11.8 | 16.8 | 16.6 | 6.3 | 16.7 |
| **Ours** | **21.0** | **8.2** | 16.2 | **14.1** | **4.7** | 16.2 |

Table 8: Experimental results of different methods on MiniGPT-4 and LLAVA-Next in the **MSCOCO-CHAIR**.

We adopt the OPOPE evaluation method proposed by HALC to assess hallucination under descriptive conditions rather than simple "yes" or "no" answers. This approach enables our method to be evaluated in a long-sentence generation environment. In practice, OPOPE employs the prompt "Please describe this image in detail" to generate captions. OPOPE then checks whether the sampled positive and negative objects appear in the generated captions to compute the POPE scores. To ensure consistency, we used the $F_{0.2}$ score, as proposed by HALC, where false negatives (FN) and the resulting recall are given less weight due to their limited trustworthiness in offline checks. Additionally, we used the same parameters and generated captions of the same average length as CHAIR.

**GPT-4V assisted evaluation:** We adopt the GPT-4V assisted evaluation method proposed by OPERA to assess the generation quality and hallucination phenomena of our approach compared to other decoding methods. Specifically, we randomly sample 500 images from the MSCOCO validation set and use decoding methods to generate descriptions for these images. The caption generation parameters and prompt we use are the same as CHAIR experiment. The evaluation involves presenting GPT-4V with the image and the corresponding descriptions generated using two decoding methods. GPT-4V is subsequently prompted to assign a score ranging from 0 to 10 for each description, evaluating two key aspects: correctness (C) and detailedness (D).

**VQA:** In Section 3, we construct a VQA task to evaluate the severity of hallucination under different resolution settings. Specifically, we select Qwen2.5-VL-7B to generate captions on the CHAIR benchmark at a resolution of 64, extract hallucinated generation samples, and substitute them into the prompt to form VQA-style questions. We further use these VQA questions to assess the logits produced in subsequent generations. To ensure that resolution settings remain meaningful, we filter images from the CHAIR benchmark such that overly low-resolution samples are excluded.

# B EXPERIMENTATION DETAILS

## B.1 EXPERIMENT SETUPS

The main generation parameters are configured as follows: the maximum number of new tokens is set to 512, top-$k$ to 5, top-$p$ to 1, and the temperature to 1. Our method targets hallucination mitigation in captions comprising multiple sentences; therefore, the maximum new tokens parameter is set to 512 to evaluate its effectiveness in long-caption scenarios. This generation length is aligned with the standard configuration in mainstream methods. The remaining parameters follow the default settings of the sampling method implemented in the HuggingFace Transformers library[1].

---

[1] https://huggingface.co/docs/transformers

## B.2 GENERATION LENGTH COMPARISON

In our experiments, similar to mainstream methods, we use 512 tokens for caption generation. Additionally, to ensure a fair comparison with other decoding methods, such as HALC and OPERA, we conduct experiments on the CHAIR benchmark with a max new tokens setting of 64, as shown in Table 7. Experimental results demonstrate that our method attains optimal performance at this generation length.

## B.3 MORE RESULTS ON CHAIR BENCHMARK

We conduct CHAIR experiments on other mainstream LVLMs, including Minigpt4 Zhu et al. (2023) and LLAVA-Next Team (2024), which are less commonly used with CHAIR compared to models such as LLAVA-1.5, Instructblip, and mPLUG-Owl2. For Minigpt4, we use Llama2 as its large language model, and for LLAVA-Next, we use the Vicuna-7B version[2]. The experimental results are shown in Table B.1. These experiments demonstrate the generalizability of our method, highlighting its ability to mitigate hallucinations even when applied to more advanced models.

## B.4 COMPREHENSIVE GPT-4V ASSISTED EVALUATION

Following the GPT-4V assisted evaluation proposed by OPERA, we conduct experiments on mainstream LVLMs such as LLAVA-1.5, InstructBLIP, and Mplug-Owl2. Two aspects are evaluated: correctness (C) and detailedness (D), both scored by GPT-4V. Since we observe that GPT-4V tends to assign higher scores to captions presented second in sequence, we construct prompts in both orders: the original prompt order, as used in OPERA's official code, where baseline captions appear first followed by our method's captions, and the reverse prompt order, where our method's captions appear first followed by baseline captions. Experimental results from Table 11 to Table 10 demonstrate that our method outperforms existing approaches in both hallucination mitigation and generation quality.

| Order | Method | InstructBLIP | | mPLUG-Owl2 | | LLAVA-1.5 | |
|---|---|---|---|---|---|---|---|
| | | C | D | C | D | C | D |
| Original Order | OPERA | 5.25 | 5.79 | 5.55 | **5.82** | 5.97 | 6.09 |
| | **Ours** | **6.02** | **6.05** | **5.56** | 5.81 | **6.03** | **6.18** |
| | Difference | +0.77 | +0.26 | +0.01 | -0.01 | +0.06 | +0.09 |
| Reverse Order | **Ours** | **5.77** | **6.02** | **6.00** | **6.5** | **6.19** | **6.4** |
| | OPERA | 5.63 | 5.70 | 5.14 | 5.58 | 5.99 | 6.39 |
| | Difference | +0.14 | +0.32 | +0.86 | +0.92 | +0.20 | +0.01 |

Table 9: Experimental results of comparing between our decoding method and OPERA decoding method on GPT4V-assist benchmark. The "original order" and "reverse order" correspond to the same content as described in Table 11.

| Order | Method | InstructBLIP | | mPLUG-Owl2 | | LLAVA-1.5 | |
|---|---|---|---|---|---|---|---|
| | | C | D | C | D | C | D |
| Original Order | HALC | 6.05 | **6.28** | **6.03** | **6.18** | 5.21 | 4.89 |
| | **Ours** | **6.14** | 6.07 | 5.97 | 6.13 | **6.13** | **6.41** |
| | Difference | +0.09 | -0.21 | -0.06 | -0.05 | +0.92 | +1.52 |
| Reverse Order | **Ours** | **6.39** | 6.14 | **6.5** | 6.19 | **6.42** | **6.38** |
| | HALC | 5.85 | **6.40** | 5.99 | **6.39** | 4.98 | 4.93 |
| | Difference | +0.54 | -0.26 | +0.51 | -0.20 | +1.44 | +1.45 |

Table 10: Experimental results of comparing between our decoding method and HALC decoding method on GPT4V-assist benchmark. The The "original order" and "reverse order" correspond to the same content as described in Table 11.

## B.5 ABLATION STUDY AND ANALYSIS

We also conduct experiment on different granularity inputs for DINO, which contains *object*, *attribute* and *relation*. The experimental results are presented in Table 14. Our analysis reveals that

---

[2]https://huggingface.co/liuhaotian/llava-v1.6-vicuna-7b

| Order | Method | InstructBLIP | | mPLUG-Owl2 | | LLAVA-1.5 | |
|---|---|---|---|---|---|---|---|
| | | C | D | C | D | C | D |
| Original Order | Greedy | 4.63 | 5.12 | 5.25 | **5.62** | 6.05 | 6.07 |
| | **Ours** | **5.73** | **5.89** | **5.63** | 5.61 | **6.14** | **6.28** |
| | Difference | +1.10 | +0.77 | +0.38 | -0.01 | +0.09 | +0.17 |
| Reverse Order | **Ours** | **6.42** | **5.96** | **6.02** | **5.87** | **6.39** | **6.4** |
| | Greedy | 4.3 | 5.09 | 4.95 | 5.8 | 5.85 | 6.14 |
| | Difference | +2.17 | +0.87 | +1.07 | +0.07 | +0.54 | +0.26 |

Table 11: Experimental results of comparing between our decoding method and greedy decoding methods on GPT4V-assist benchmark in OPERA paper. The "original order" refers to the prompt where greedy captions appear first, followed by our method's captions. In contrast, the "reverse order" refers to the prompt where our method's captions appear first, followed by greedy captions.

DINO struggles to effectively localize *attribute* and *relation*, resulting in excessive meaningless grounding.

**Equation 3 – Why use *minimum* instead of *average***: We conducted an ablation study comparing the use of minimum versus average CLIP similarity scores in Equation 3. As shown in Table below, the performance difference is modest, but the minimum yields slightly better hallucination reduction. We adopt the minimum aggregation as it provides a more conservative and principled estimate by emphasizing the least-aligned region, which is important for flagging potential hallucinations.

| Aggregation | $CHAIR_S \downarrow$ | $CHAIR_I \downarrow$ |
|---|---|---|
| average | 39.4 | 11.6 |
| minimum | **38.0** | **11.4** |

Table 12: Ablation study on the aggregation strategy in Equation 3. Using the *minimum* CLIP score slightly improves hallucination reduction by focusing on the weakest region-text alignment.

| | $CHAIR_S \downarrow$ | $CHAIR_I \downarrow$ |
|---|---|---|
| $\sigma$ (original) | 30.8 | 11.4 |
| $C = 0.1$ | 31.1 | 11.2 |
| $C = 1$ | **24.8** | 6.8 |
| $C = 10$ | 25.0 | **6.7** |

Table 13: Ablation study on hyperparameter $C$ in Equation 7. The best hallucination mitigation is achieved when $C = 1$, which we adopt as default.

We also observe that replacing $\sigma$ with a constant $C$ reduces the influence of potentially noisy visual signals, and in some cases leads to improved stability. This supports the use of calibrated visual guidance for more robust alignment.

## C  TIME ANALYSIS.

Figure 5 demonstrates that the best results are achieved with a sampling time of 3. To optimize generation efficiency, we set the sampling time to 3 for all experiments. Table 15. The experimental parameters for each method are selected based on their best performance. The results indicate that our method achieves state-of-the-art hallucination mitigation while maintaining competitive generation efficiency.

Based on the Biber et al. (2000), nouns comprise approximately 25% of generated words. Since sentence-level decoding is independent, these steps can be **parallelized**, enabling a tractable estimation of time cost. Assuming an average sentence length of $m$ words, and that each sentence triggers one additional CLIP evaluation, the average per-token time cost can be approximated as:

$$T_{\text{LVLM}} + 0.25 \times (T_{\text{DINO}} + T_{\text{CLIP}}) + \frac{1}{m}T_{\text{CLIP}}$$

Here, $T_{\text{LVLM}}$ denotes the time required for the base vision-language model to decode one token. The term $0.25 \times (T_{\text{DINO}} + T_{\text{CLIP}})$ reflects the fact that roughly 25% of tokens (nouns) are grounded

| Greedy | Rel | Attr | Obj | $CHAIR_S$ | $CHAIR_I$ |
|--------|-----|------|-----|-----------|-----------|
| ✓ | | | | 47.0 | 13.6 |
| | ✓ | | | 47.4 | 13.6 |
| | | ✓ | | 46.6 | 13.9 |
| | | | ✓ | **22.2** | **5.8** |

Table 14: Comparison of $CHAIR_S$ and $CHAIR_I$ for different DINO inputs, **Bold** values represent the best results.

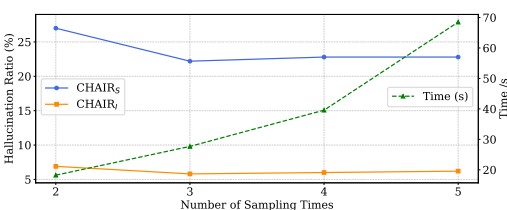

Figure 5: Performance of our method sampling times in range of 2 to 5

using DINO and undergo additional CLIP validation, while the $\frac{1}{m}T_{\text{CLIP}}$ accounts for sentence-level scoring applied once per sentence.

In practice, since $T_{\text{DINO}}$ and $T_{\text{CLIP}}$ are significantly smaller than $T_{\text{LVLM}}$, the overall time cost is close to standard greedy decoding. Therefore, despite the integration of two alignment modules, the expected runtime overhead remains minimal due to both their low per-call latency and the parallelizable nature of the added operations.

## D ALGORITHM

We summarize the process of our method in Algorithm 1.

---
**Algorithm 1** Our method's Algorithm

---
**Input:** LVLM parameterized by $\theta$, sampling times $k$, candidates number $N$, weight hyperparameter $\alpha$, image input $x_{img}$ and text prompt $s_0$
**Parameter:** $\theta$, $k$, $N$, $\alpha$
**Output:** $y$
1: Let $t = 0$
2: $\text{SET}_0 \leftarrow \{\langle \text{Input}(\text{x}_{img}, s_0) \rangle\}$
3: **while** $\text{SET}_t$ is not empty **do**
4:     $\text{SET}_{t+1} \leftarrow \emptyset$
5:     **for all** candidate in $\text{SET}_i$ **do**
6:         **repeat**
7:             Sample $s \sim \text{LVLM}_\theta(s_{t+1}|x, s_0, s_1, \ldots, s_{ij})$
8:             $F(x_{\text{img}}, s_{ij}) = (1-\alpha)\big(g_{\text{DINO}}(x_{\text{img}}, s_{ij}) + g_{\text{CLIP}}(x_{\text{img}}, s_{ij})\big) + \alpha \cdot f_\theta(s_{ij})$
9:             $\text{SET}_{t+1} \leftarrow \text{SET}_{t+1} \cup \{x, s_0, s_1, \ldots, s_{ij}\}$
10:         **until** $k$ times
11:     **end for**
12:     Rank $\text{SET}_{t+1}$ by F(s)
13:     $\text{SET}_{t+1} \leftarrow$ Top N candidates in $\text{SET}_{t+1}$
14:     $t \leftarrow t + 1$
15: **end while**
16: $y = \arg\max(\text{SET})$
17: **return** $y$

---

## E EFFECTIVENESS OF OUR FRAMEWORK IN OTHER DOMAINS

We further evaluate our method on the ECSO task ("Eyes Closed, Safety On": Protecting Multimodal LLMs via Image-to-Text Transformation, ECCV 2024) to assess its generalizability. As shown in Figure 16, our approach not only reduces hallucinations but also improves the overall

| Method | Decoding Time |
|--------|---------------|
| Greedy | **3.90** |
| HALC | 89.88 |
| CGD | 19.13 |
| Ours | 27.68 |

Table 15: Comparison of time cost of different decoding method. The parameters are configured to the official settings.

quality of generated descriptions. These results highlight the broader applicability of our framework beyond hallucination-specific benchmarks.

| Backbone: LLAVA-1.5 | Baseline | ECSO | VCD | Ours |
|---------------------|----------|------|-----|------|
| **SD** | 82.95 | 89.17 | 89.21 | **92.32** |
| **OCR** | 59.44 | 79.45 | 79.35 | **90.50** |
| **SD_OCR** | 59.73 | 77.69 | 77.50 | **88.73** |

Table 16: Performance comparison on MM-SafetyBench . Baseline denotes LLaVA-1.5 without any enhancement. ECSO is a prior method. VCD uses VCD decoding as an improvement. Ours is our proposed method.

# F CASE STUDY

We present case studies on hallucination mitigation using the COCO 2014 validation dataset, as shown in Figure 6. The table compares captions generated by Greedy, HALC, CGD, and our proposed approach for the images in the leftmost column. Notably, our method generates longer and more detailed captions. Hallucinated content in the descriptions is highlighted in red.

| Image | Greedy | HALC | CGD | Ours |
|---|---|---|---|---|
|  | The image features a delicious breakfast plate filled with various fruits and waffles. The plate is placed on a dining table, and it contains a generous portion of blueberries, apples, and bananas. The fruits are spread across the plate, with some blueberries on the left. | The image features a delicious breakfast plate filled with various fruits and waffles. The plate is placed on a dining tablecloth, and it is adorned with a blue floral pattern. The breakfast consists of two waffles, each topped with blueberries and apples. | The image shows a plate filled with delicious food consisting of pancakes topped with sliced apples, blueberries, and syrup. The blueberries are spread out across the plate, with some closer to the center and others closer to the edges. This combination of flavors is not only visually appealing but also looks like a tasty and healthy breakfast option. | The image displays a plate of delicious food, containing waffles, fruit, and syrup on a dining table. The waffles are garnished with fresh blueberries, adding a touch of color and flavor to the dish. There are multiple blueberries on the plate, distributed evenly throughout the waffles. The presentation of this scrumptious breakfast is quite appetizing and inviting. |
|  | The image features a woman with short hair, wearing a black shirt, and holding a banana in her mouth. She appears to be making a funny face while eating the banana. The woman is standing in a kitchen, with a refrigerator visible in the background. | The image features a woman with short hair, wearing a black shirt, and holding a banana in her mouth. She appears to be making a funny face while eating the banana. The woman is standing in a kitchen, with a refrigerator visible in the background. | In the image, a woman is sitting in a kitchen, chewing on a banana. She is leaning back with the banana in her mouth, and her eyes are closed. In the background, there is an oven and a microwave visible, which are likely part of the kitchen appliances. Additionally, a knife can be seen placed nearby on the counter. | She is wearing a black shirt and is holding a half-eaten banana in her mouth. The woman is looking at the camera, showcasing her unique style and humor. In the background, there are a few kitchen items visible. An oven is placed near the top left corn is placed near the top left corner of the scene, and a cup is located on the right side. |
|  | The image depicts a small, cluttered kitchen with a wooden dining table surrounded by chairs. The table is covered with papers, and there are several chairs placed around it. A refrigerator is located on the left side of the room, and a microwave can be seen above. | The image depicts a small, cluttered kitchen with a wooden dining table surrounded by chairs. The table is covered with papers, and there are several chairs placed around it. A person is standing near the table, possibly working on the papers. | The image depicts a large residential garage turned into a multipurpose room, featuring various appliances and furniture. There is a refrigerator on the left side of the room, surrounded by numerous chairs and dining tables. Some of the chairs are closer to the refrigerator, while others are situated around the dining tables. | The image depicts a small home office space, likely an unfinished garage, set up for various activities. It is equipped with a refrigerator situated on the left side of the room and a white table in the center. The table is surrounded by four wooden chairs, with two chairs positioned on the left side, one on the right side, and one chair at the head of the table. A variety of objects and supplies can be found throughout the space. |
|  | The image features a woman standing in front of a video game display, possibly at a convention or a store. She is wearing a white shirt and appears to be observing the game. The display consists of a large TV screen and a Wii console, with a remote control placed nearby. A chair is positioned in the background, and a handbag can be seen placed on the floor. | The image features a woman standing in front of a video game display, possibly at a convention or a store. She is wearing a white shirt and appears to be observing the game. The display consists of a large TV screen and a Wii console, with a man's image on the screen. There are also a few chairs in the area, with one located near the center of the scene and another towards the right side. A chair is also present in the background. | The scene features a person standing in front of a video game display, which includes a Nintendo Wii gaming console with a TV screen attached. The display is set up in a booth-like area to attract visitors, and there is a person positioned in the background of the display, potentially working behind the counter. A chair is positioned in the background, and a handbag can be seen placed on the floor. | The image displays a busy event featuring a large screen in the center, which appears to be a Nintendo Wii game. Numerous individuals can be seen playing games at the event, with some standing around and enjoying the experience. The main display features a black and white image of a man playing with a Nintendo Wii, likely on a television screen or a large monitor. A row of figures, representing the Wii players, are also present, likely set up on the front of the screen for an interactive element at the event. |

Figure 6: A comparison of text generated by Greedy Search, HALC, CGD, and our proposed method, using examples from the COCO 2014 validation dataset with LLaVA-1.5. The hallucinated parts are highlighted in red.

