# OpenReview forum: "Mitigating Object Hallucinations in Large Vision-Language Models via multi-scale visual integration"
_ICLR.cc/2026/Conference — Submitted to ICLR 2026_

### Official Review · Reviewer_Di87 · 2025-10-25

**Soundness:** 2
**Presentation:** 2
**Contribution:** 3
**Rating:** 4
**Confidence:** 3

**Summary:**

The paper analyzes object hallucination in LVLMs through a resolution-scaling ablation study. It reports a boundary effect: moderately higher visual resolution reduces hallucinations, but pushing resolution too far increases visual tokens, diffuses attention, and brings hallucinations. Building on this, the authors propose a multi-scale visual decoding framework that mixes (i) high-res object-level grounding and (ii) lower-res global attention constraints, with a fusion-based score that combines these signals with model likelihood to steer decoding. Experiments on CHAIR/POPE and a GPT-4V-assisted evaluation show improvements over several contrastive decoding baselines.

**Strengths:**

1. To the best of my knowledge, this is the first paper to analyze hallucination as a function of image resolution. I agree with the central claim: for models relying on discrete tokenizers, overly high resolution can harm input consistency and, in turn, confuse the VLM.

2. As a remedy, using Grounding DINO on high-resolution inputs for fine-grained consistency, while using CLIP on low-resolution inputs for global alignment, is a reasonable high-level design.

**Weaknesses:**

1. (Major) The paper’s motivation is under-analyzed. It only shows performance curves across resolutions without digging into root causes that can suggest logical reasons for the proposed solution. Because this logical chain from motivation to proposed methods is very thin, I remain unconvinced that these two specific components (Grounding DINO + CLIP) are the right answer. Although Table 5 provides an ablation study, it shows only modest quantitative changes and does not substantiate the necessity of the method.

2. (Major) The approach ultimately depends on external models, which makes direct comparison to purely internal (decoding-only) baselines difficult. (At least, the same computational costs, wall-clock latency or FLOPs, etc..)

3. (Moderate) The text claims the overhead is “close to greedy,” yet the timing table reports 27.7s vs 3.9s for greedy (and ~19.1s for CGD), which is a substantial slowdown for captioning-length outputs. This gap matters for interactive LVLM use. A tighter runtime analysis—or an anytime / early-exit variant—would help.

**Questions:**

1. Why didn’t you analyze the root causes of resolution-induced hallucination more deeply? Personally, I suspect a primary factor is that a discrete tokenizer receives different semantic units as resolution changes, but there could be multiple contributing reasons. The solution should be derived from such a causal analysis, yet the current connection feels too weak.

2. Why specifically choose Grounding DINO and CLIP? I don’t see a compelling rationale for why these particular methods are necessary.

---

> ### Author Response · Authors · 2025-11-28
>
> ## How Our Method Approximates the Theoretical Ideal High-Resolution Attention Function
>
> ### Theoretical Ideal Attention Function
>
> A hallucination-reduced LVLM operating at high resolution relies on an ideal attention function:
>
> $F^\*(x_{\text{img}}, s_{ij}) = \text{Attend}(t_i, v_j)$
>
> which satisfies two fundamental properties:
>
> ---
>
> ### 1. Suppressing irrelevant or redundant tokens
>
> For irrelevant visual tokens, the ideal function satisfies:
>
> $\text{Attend}(t_i, v_j) \approx 0$
>
> meaning that non-grounding regions should receive negligible attention.
>
> ---
>
> ### 2. Concentrating attention on correct object regions
>
> For relevant visual regions:
>
> $\text{Attend}(t_i, v_j)$ is sharply concentrated on the correct object regions even at high resolution.
>
> This ensures that increasing resolution improves grounding instead of introducing noise.
>
> ---
>
> ## Practical Problem: Real LVLMs Fail to Achieve the Ideal Function
>
> As shown in Section 3, current LVLMs do not realize these ideal attention behaviors.
> Increasing resolution introduces redundant background tokens, resulting in:
>
> - diffused attention,
> - weakened grounding,
> - increased hallucination.
>
> This motivates approximating the ideal attention function $F^\*$.
>
> ---
>
> ## Our Approach: Approximating the Oracle Attention via Dual Constraints
>
> We approximate the ideal behavior through two complementary supervision paths:
>
> - fine-grained grounding (GroundingDINO),
> - global semantic alignment (CLIP).
>
> Our final scoring function is:
>
> $F(x_{\text{img}}, s_{ij}) = (1 - \alpha)\(G(x_{\text{img}}, s_{ij}) + A(x_{\text{img}}, s_{ij})) + \alpha \cdot f_\theta(s_{ij})$
>
> where:
>
> - $G(\cdot)$ gives region-level grounding,
> - $A(\cdot)$ gives global alignment,
> - $f_\theta(\cdot)$ gives the decoder's internal score.
>
> ---
>
> ## How the Two Paths Approximate the Ideal Attention Function
>
> ### 1. Approximating the ideal suppression behavior: $\text{Attend}(t_i, v_j) \approx 0$
>
> - CLIP assigns low alignment scores to mismatched image–text pairs.
> - This lowers the final score $F(x_{\text{img}}, s_{ij})$, suppressing irrelevant or hallucinated content.
>
> ---
>
> ### 2. Approximating the ideal concentration on correct regions
>
> - GroundingDINO provides strong region-level grounding at high resolution.
> - Correct object regions receive high grounding scores $G(x_{\text{img}}, s_{ij})$.
>
> Thus:
>
> $\text{Attend}(t_i, v_j)$ becomes large only in the correct visual regions.
>
>
> ## Combined Approximation
>
> Together, the two supervision paths approximate the oracle behavior:
>
> $Attend(t_i, v_j) \approx Attend_{local}(t_i, v_j)\cap Attend_{global}(t_i, v_j) = G(x_{img}, s_{ij}) + A(x_{img}, s_{ij})$
>
>
> providing a practically attainable approximation to the ideal attention function in modern LVLMs.
>
> ---
>
> ## Q2: Follow-up on Computational Efficiency
>
> As discussed in Appendix C, our decoding framework is theoretically parallelizable across tokens, enabling latency close to greedy decoding when parallel execution is used. Although the current implementation is not fully optimized, further efficiency improvements remain an important direction for future work.

---

### Official Review · Reviewer_Jpiz · 2025-11-01

**Soundness:** 3
**Presentation:** 2
**Contribution:** 3
**Rating:** 6
**Confidence:** 4

**Summary:**

This study proposes a method that reduces the hallucination problem of LVLMs by considering both low and high resolution visual input tokens.

The authors are first inspired by the fact that the sampling distribution gap between LVLM with and without image input peaks at intermediate resolution. Based on that , they discovered the phenomenon that the attention distribution varies significantly with different resolution setup, while different setup actually contributes to answering different level of questions.

To utilize this observation, this study proposed a method that constrains the attention mechanism for both high-resolution and low-resolution settings. Experiments show that the proposed method can significantly reduce the hallucination problem of LVLMs on different tasks.

**Strengths:**

see summary

**Weaknesses:**

The weakness is mainly in presentation. I think Figure 2 taking half page does not make sense and Figure 4 is not expressive enough.

**Questions:**

n/a

---

> ### Author Response · Authors · 2025-11-28
>
> We appreciate the reviewer’s feedback regarding presentation clarity. We will refine the visual design and layout in the revised version—e.g., reducing the footprint of Figure 2 and improving the clarity and expressiveness of Figure 4—to ensure they convey the intended information more effectively.

---

### Official Review · Reviewer_2ni9 · 2025-11-01

**Soundness:** 2
**Presentation:** 1
**Contribution:** 2
**Rating:** 2
**Confidence:** 4

**Summary:**

This paper addresses the issue of object hallucination in large vision-language models (LVLMs), where models generate descriptions of non-existent objects or details. The authors identify a boundary effect in visual resolution scaling: while moderate increases in resolution reduce hallucination, excessively high resolutions lead to attention diffusion, causing hallucinations to reemerge.

To overcome this, the authors propose a multi-scale visual decoding framework that integrates: fine-grained grounding via GroundingDINO to recover object-level details, and global attention constraints via CLIP to maintain semantic alignment. These are combined with the LVLM’s internal likelihood in a fusion module to guide decoding. The method is training-free and does not require external data.

Extensive experiments on CHAIR, POPE, and GPT-4V-assisted evaluations demonstrate state-of-the-art performance in reducing hallucinations while maintaining or improving caption quality across multiple LVLM backbones (e.g., LLaVA-1.5, InstructBLIP, mPLUG-Owl2). Ablation studies confirm the contribution of each component, and the method shows promising generalizability to safety-critical domains.

**Strengths:**

While many researchers advocate for simply increasing image resolution to mitigate hallucinations, this work demonstrates the existence of a "sweet spot." It shows that beyond this point, further increasing resolution can paradoxically exacerbate hallucinations due to "attention diffusion." This aligns with the intuition that an excessive number of image tokens, coupled with the VLM's inherent tendency to attend less to visual tokens as text generation progresses, can degrade performance. The paper's strength lies in directly addressing this specific, well-motivated problem.

The method's strength lies in its design, presenting an innovative "plug-and-play" solution. The core of the method is a training-free, multi-scale visual decoding framework. Its ingenuity is demonstrated through the creative and purposeful combination of existing powerful tools (GroundingDINO and CLIP), rather than the introduction of complex new modules. The fine-grained grounding module accurately captures object-level evidence at high resolution, while the global attention constraint module maintains semantic consistency at a lower resolution; together, they synergistically address the "attention diffusion" paradox caused by simply increasing resolution. Furthermore, the method employs a carefully designed fusion scoring function that organically integrates visual grounding, semantic alignment, and the model's internal likelihood, supplemented by an iterative decoding strategy. This ensures that the output remains faithful to the image content while respecting the generative capabilities of the underlying model. This design allows the framework to be directly applied to a variety of existing LVLMs demonstrating exceptional practicality and generality.

**Weaknesses:**

The computational efficiency of the proposed method is a significant concern, potentially more central and severe than presented in the limitations section. The core approach, which relies on two external models (GroundingDINO and CLIP) and employs iterative decoding, inherently introduces substantial computational overhead. As shown in Appendix Table 15, the decoding time of the proposed method (27.68s) is substantially higher than standard greedy decoding (3.90s) and is even slower than some existing methods (e.g., CGD at 19.13s). This severely limits its practicality for real-time or large-scale applications.

While the experiments are comprehensive on standard benchmarks like MS-COCO, these datasets are relatively "clean." The paper's limitation regarding performance in "densely packed objects" or "specialized domains," while valid, could be more concretely addressed. The generalizability and robustness of the method would be significantly strengthened by evaluating it on more challenging benchmarks, such as those involving multi-turn VQA, which require sustained visual grounding over a conversational context.

For experiments such as on CHAIR, there is no sensitivity/statistical significance analysis showing the randomness of the results, as we know the scores could vary with a large variance. The paper could benefit from testing on more recent benchmarks and models, such as InternVL, PaliGemma, Phi vision, Meta PLM, AMBER, MME, THRONE.

Paper could benefit from better writing and cleaner presentation.

**Questions:**

The core analysis in Section 2, which establishes the "boundary effect" of resolution scaling, is conducted using Qwen2.5-VL.  I remember that Qwen2.5-VL's visual encoder has a maximum input resolution (e.g., 448x448 pixels, can you check the specific number ).  For images larger than this limit, the standard practice is for the visual processor to downsample them to the model's acceptable size.

Could you please clarify in detail how you handled the input images for the high-resolution settings reported in your experiments (e.g., 896 as shown in Figure 1a)?

Specifically, were the images at these high resolutions (e.g., 896x896) fed directly to the Qwen2.5-VL model, which would then internally downsample them to its maximum capacity?

Or, did you employ a different preprocessing strategy (e.g., patch-based processing) that truly allows the model to ingest and tokenize the full high-resolution image?
This clarification is crucial because: If the images were downsampled, the observed performance degradation at high resolutions might not be due to "attention diffusion" from an excessive number of visual tokens, but rather an artifact of information loss or distortion caused by aggressive resizing.  This would fundamentally challenge the proposed hypothesis. A clear explanation of the input pipeline is necessary to validate that the model indeed received and processed the intended long visual token sequences, which is the cornerstone of your analysis.

---

> ### Author Response · Authors · 2025-11-28
>
> Q1: Computational Efficiency
> We appreciate the reviewer’s insightful comments regarding computational efficiency. As analyzed in Appendix C, our decoding framework is theoretically parallelizable—each refinement step operates independently across sentences—implying that its asymptotic decoding latency can approach that of standard greedy decoding when parallel execution is enabled.
> In practice, our initial implementation was not fully optimized, which resulted in the higher wall-clock time reported in Table 15. We will include updated measurements in the revised version. We acknowledge that further improvements are possible and consider efficiency optimization an important direction for future work.
>
> Q2: Evaluation Beyond “Clean” Datasets
> We acknowledge the reviewer’s concern regarding evaluation scope. As stated in the limitation section, although our method was not exhaustively tested across all specialized domains, we conducted experiments on safety-critical benchmarks, which are substantially more challenging and domain-specific. These results demonstrate that our approach generalizes effectively beyond standard vision benchmarks.
> Specifically, within the ECSO evaluation benchmark, we used our method to generate safety-aligned captions and achieved performance surpassing ECSO’s reported harmless rate. Detailed results are provided in Appendix E. We believe these findings offer strong evidence of robustness and generalizability in domain-specific reasoning scenarios.
>
> Q3: Applicability to VQA-Style Benchmarks
> As noted in Lines 668–670, our method is not designed for one-word response tasks such as VQA, where the output format fundamentally differs from our refinement-based generation framework. Consequently, benchmarks like MME or AMBER—which heavily rely on one-word answers—are not directly applicable to our method.
> Regarding broader validation, we agree with the reviewer and have already included experiments on LLaVA-Next, a recent and strong vision-language model. The results in Appendix B.3 show consistent compatibility and effectiveness. We appreciate the suggestion and plan to extend evaluations to additional emerging models in future work.
>
> Additional Clarification: High-Resolution Input for Qwen2.5-VL
> We thank the reviewer for raising this important clarification request. The concern is based on the assumption that Qwen2.5-VL uses a fixed low-resolution encoder (e.g., 448×448) and therefore must downsample high-resolution inputs. However, Qwen2.5-VL adopts a dynamic-resolution visual encoder, and—consistent with Qwen2-VL—the technical report does not specify a 448×448 resolution cap; instead, it supports substantially higher resolutions (up to ~3.5k×3.5k).
> In our experiments, the high-resolution settings (e.g., 768 in Fig. 1(a)) fall well within this range. For a target resolution **R** (e.g., 448, 672, 768), we resize the input image such that its shorter side equals **R** (while preserving aspect ratio) and directly feed it into Qwen2.5-VL. No external tiling or forced downsampling to 448×448 is performed; the model internally tokenizes the full high-resolution image under its native dynamic-resolution mechanism.
> Therefore, the performance degradation at high resolutions is not due to aggressive resizing artifacts but rather reflects the model’s processing of longer visual token sequences—precisely the behavior underpinning our “boundary effect” analysis. We will clarify this pipeline more explicitly in the revised version.

---

### Meta-Review · Area_Chair_DtDd · 2025-12-22

**Summary:**

This paper addresses the persistent challenge of object hallucination in Large Vision-Language Models (LVLMs). The authors introduce the concept of a "resolution boundary effect," observing that while increasing visual resolution initially aids performance, excessive resolution leads to "attention diffusion," paradoxically reintroducing hallucinations. To counter this, the paper proposes a training-free, multi-scale visual alignment decoding framework. This method integrates fine-grained grounding via GroundingDINO (at high resolution) and global semantic constraints via CLIP (at low resolution), fusing these with the model's internal logits to guide generation.

Reviewers acknowledged the novelty of the "resolution boundary effect" insight and the intuitive logic of combining global and local visual constraints. However, significant concerns regarding computational efficiency, reliance on heavy external models, and the depth of the motivational analysis were raised. Despite clarifications regarding the input processing of Qwen2.5-VL during the rebuttal, the fundamental issues regarding the method's practicality and design justification remain. Therefore, the Area Chair (AC) recommends rejection.

**Reviewer Concerns:**

- Prohibitive Computational Cost: Both Reviewers 2ni9 and Di87 flagged the severe latency overhead as a critical flaw. The proposed method takes ~27.7s per sample compared to 3.9s for greedy decoding, making it nearly 7x slower. While the authors argued in the rebuttal that the method is "theoretically parallelizable," reviewers found the current implementation's reliance on iterative decoding and two external models (GroundingDINO and CLIP) to be too heavy for practical "plug-and-play" deployment.
- Weak Motivation for Specific Architecture: Reviewer Di87 criticized the "thin" logical chain between the observed problem (resolution-induced hallucination) and the chosen solution. The reviewer argued that simply stacking GroundingDINO and CLIP feels like an engineering heuristic rather than a principled fix derived from a deep causal analysis of *why* discrete tokenizers fail at high resolutions.
- Unfair Comparisons & External Dependencies: Reviewer Di87 noted that relying on external vision experts makes direct comparison with internal decoding-only baselines unfair, as the proposed method has access to significantly more encoded information and compute.

**Reviewer Scores:**

- Practicality vs. Performance: Reviewers 2ni9 (Score 2) and Di87 (Score 4) were not swayed by the performance gains because the cost (latency and complexity) is disproportionately high. The rebuttal's promise of future optimization did not address the immediate concern that the method is currently impractical for real-time applications.
- Validity of the Solution: The skepticism from Reviewer Di87 regarding the *necessity* of the specific GroundingDINO+CLIP design remains. Without a stronger ablation proving that this specific combination is the *only* or *optimal* way to solve the resolution paradox (rather than just reducing hallucination via brute-force external grounding), the contribution is seen as limited.

---

### Decision · Program_Chairs · 2026-01-26

Reject